# Heterogeneous encoding of temporal stimuli in the cerebellar cortex

Chris. I. De Zeeuw [1,2,3], Julius Koppen[1,3], George. G. Bregman[1], Marit Runge[1] & Devika Narain [1] ✉

Local feedforward and recurrent connectivity are rife in the frontal areas of the cerebral cortex, which gives rise to rich heterogeneous dynamics observed in such areas. Recently, similar local connectivity motifs have been discovered among Purkinje and molecular layer interneurons of the cerebellar cortex, however, task-related activity in these neurons has often been associated with relatively simple facilitation and suppression dynamics. Here, we show that the rodent cerebellar cortex supports heterogeneity in task-related neuronal activity at a scale similar to the cerebral cortex. We provide a computational model that inculcates recent anatomical insights into local microcircuit motifs to show the putative basis for such heterogeneity. We also use cell-type specific chronic viral lesions to establish the involvement of cerebellar lobules in associative learning behaviors. Functional heterogeneity in neuronal profiles may not merely be the remit of the associative cerebral cortex, similar principles may be at play in subcortical areas, even those with seemingly crystalline and homogenous cytoarchitectures like the cerebellum.

The cerebellum, a neural structure whose origins can be traced back to primitive marine species, has maintained the primary histological features of its cortical wiring throughout vertebrate evolution[1]. The contiguous lattice-like construction[2] of the cerebellar cortex has inspired several theories that expect such structural uniformity to be translated into homogeneous physiological function[3,4]. Recent work, however, has revealed new forms of heterogeneity that belie the crystalline cytoarchitecture of the cerebellar cortex. Evidence from different species and behaviors suggests that the cerebellum exhibits heterogeneity of various forms, including diversity in transcriptomic identity[5], microcircuit connectivity[6–8], intrinsic cellular properties[9], temporal characteristics[10], synaptic diversity[11], and diverse task-related functions[12].

It is, however, challenging to reconcile these newfound sources of heterogeneity with the fact that for the past several decades, physiological responses of primary neurons in the cerebellar cortex, the Purkinje cells, have often comprised relatively simple forms of activity, i.e., patterns of increase, decrease, and, during periodic behaviors, oscillations between these two states. Such facilitation and suppression of cerebellar cortical neurons have been reported in a variety of behaviors, such as smooth pursuit[13–16], motor control[17], vestibular-ocular responses[18,19], ocular following[20], manual tracking[21], eyelid conditioning[22,23], whisker control[24], saccadic adaptation[25,26] and manual reaching behaviors[27]. Recently, more varied but relatively simple functional patterns have been uncovered in cerebellar neurons encoding body movements[28], suggesting that cerebellar encoding may support diverse representations. One might argue that the overall lack of physiological heterogeneity is unsurprising given that the cerebellar cortex is considered a feedforward structure with only limited potential for recurrent dynamics but this view has recently been called into question[29,30]. This change of opinion, in part, arises from the revelation of the surprising degree to which local recurrent and feedforward collateralization pervades local cerebellar cortical microcircuits leading to interconnected anatomical and functional motifs[6–8]. These findings, alongside diversity in incoming extracerebellar inputs, predict more complex dynamics in cerebellar cortical activity than currently reported.

---

[1]Department of Neuroscience, Erasmus University Medical Center, Rotterdam, The Netherlands. [2]Netherlands Institute of Neuroscience, Amsterdam, The Netherlands. [3]These authors contributed equally: Chris. I. De Zeeuw, Julius Koppen. ✉e-mail: d.narain@erasmusmc.nl

We know that recurrent connectivity in frontal areas of the cerebral cortex[31] gives rise to complex, persistent, and heterogeneous dynamics[32,33], which often feature multiplexed task representations[32–36]. The question arises, given recently uncovered local recurrent and feedforward microcircuit motifs and given input diversity to various cerebellar lobules, why such heterogeneity eludes cerebellar cortical activity underlying behavioral tasks. Here, we show evidence of multiplexed task-related representations and functional heterogeneity in cerebellar cortical responses in a relatively simple associative learning task. We provide computational models of the cerebellar microcircuit that incorporate recent anatomical insights[6,8] to predict how such distinct and diverse representations might be acquired and propagated within the cerebellar cortex.

## Results

### Heterogeneity of cerebellar cortical responses in temporal association behaviors

We study predictive eyelid responses in a modified trace conditioning task, where mice were trained on different statistical distributions of time intervals (Fig. 1a, b). Trials consisted of a transient flash of light paired with the delayed administration of a periocular airpuff. The interstimulus interval (ISI) between the light and airpuff was either a single interval (350 ms) referred to as a 'Narrow' condition or was drawn from a discrete uniform distribution (200, 275, 350, 425, 500 ms), referred to as a 'Wide' condition (Fig. 1b). After several pairings, the eye would close predictively (Fig. 1c–e) at the anticipated time of the airpuff. These predictive responses are also known as conditioned responses (CR), whereas reflexive eyelid closure in response to the airpuff is known as the unconditioned response. $A_{CR}$ refers to the amplitude of the conditioned response, which was best evaluated on test trials, where the airpuff was omitted (Fig. 1b–d below and Supplementary fig. 1a). After training each mouse for several weeks (average training time ~70 days), we found that predictive eyelid responses stabilized in amplitude, $A_{CR}$, and frequency (CR percentage)

(Supplementary fig. 1b). We examined the predictive eyelid response across test trials for mice trained on Narrow and Wide conditions and found them to differ in shape and amplitude (Fig. 1d, e and Supplementary fig. 1a).

After the stabilization of responses, we performed acute large-scale electrophysiological recordings of neurons in the cerebellar lobules IV/V and Simplex across days (Fig. 2a) in individual mice that were trained on the Narrow or Wide conditions. Acute recordings at the same location across days were enabled by a grid-in-chamber apparatus that ensured that probe locations were within close proximity on each acute penetration (Supplementary fig. 2c). Subsequent histological registration and analysis of probe data confirmed that the vast majority of recordings were performed in lobules IV/V and Simplex (Supplementary fig. 3a, b).

We identified Purkinje cells using cross-correlation statistics between Purkinje cell (PC) simple and complex spiking patterns (Supplementary fig. 4a, b, see "Methods" section). Else, units were classified as putative Purkinje cells (pPC) and putative molecular layer interneurons (pMLI) by considering other factors (see "Methods" section for details). We analyzed these cerebellar cortical neurons recorded during the Narrow ($N_{mice} = 6$, $N_{Neurons} = 2571$) and Wide ($N_{mice} = 6$, $N_{Neurons} = 2179$) conditions and found heterogeneity in neural activity responses, not unlike the diversity of complex, multiplexed responses often reported in the cerebral cortex[32,33,35]. We found that these activity patterns could be classified into at least eight distinct functional classes (Figs. 2b–e and 3a–g, and Supplementary fig. 5a, b). The classification algorithms measured modulation in the stimulus Interval epoch (I) and the epoch following the Airpuff (A) and took the valence and modulation of activity following both epochs into consideration (Figs. 2b–d and 3a–g, and Supplementary fig. 5a, b).

We found that a large number of neurons modulated congruently for the two epochs, i.e., I+A+ or I−A−, in both Narrow (54%) and Wide (41%) conditions, whereas a smaller proportion of neurons exhibited incongruent modulation, i.e. I−A+ or I+A−, for the Narrow (7%) and

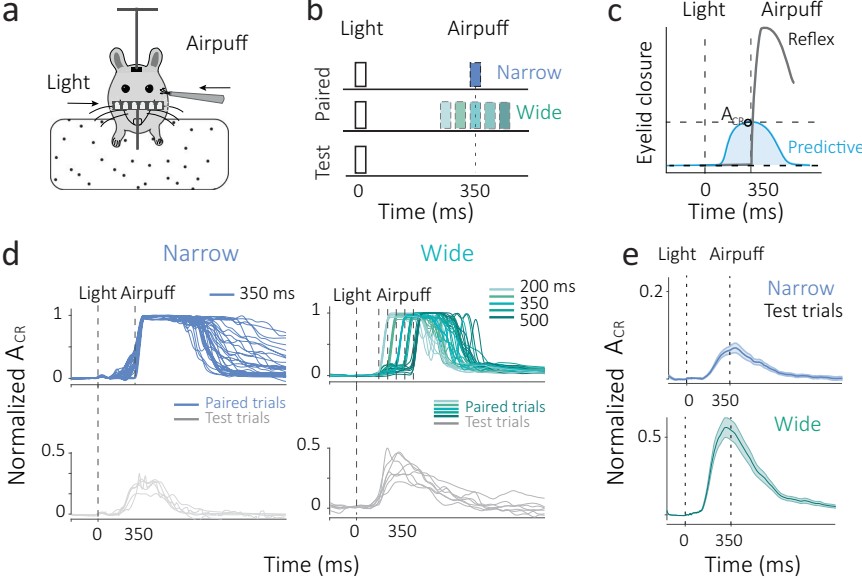

**Fig. 1 | Task and behavior. a**, **b** Mice are trained on a modified trace eyeblink conditioning task (left) where the inter-stimulus time interval (ISI) between a flash of light and a transient periocular airpuff on a paired trial is sampled from a 'Narrow distribution' with a single ISI (350 ms, blue, top) or sampled from a discrete uniform 'Wide distribution' (200, 275, 350, 425, 500 ms, Green, middle). The airpuff is omitted in test trials (right, bottom panel) to observe the predictive component of the eyeblink response. **c** Eyelid closure is measured on test and paired trials to determine the amplitude ($A_{CR}$) of the predictive response (blue) and distinguish it from the reflexive response (gray) at the time of the airpuff. **d** Normalized single-trial eyelid closure traces for paired (above, blue/green lines) and test trials (below, gray lines) for the Narrow and Wide conditions. The traces are aligned to the onset time of the light cue (black dashed line) and the time of the airpuff(s) is also indicated (black dashed lines). **e** Examples of average normalized eyelid traces for test trials in the Narrow (blue) and Wide (green) conditions. Note that no airpuff was administered in these trials but the mean of both interval conditions is indicated (black dashed line at 350 ms). Error bars indicate standard error.

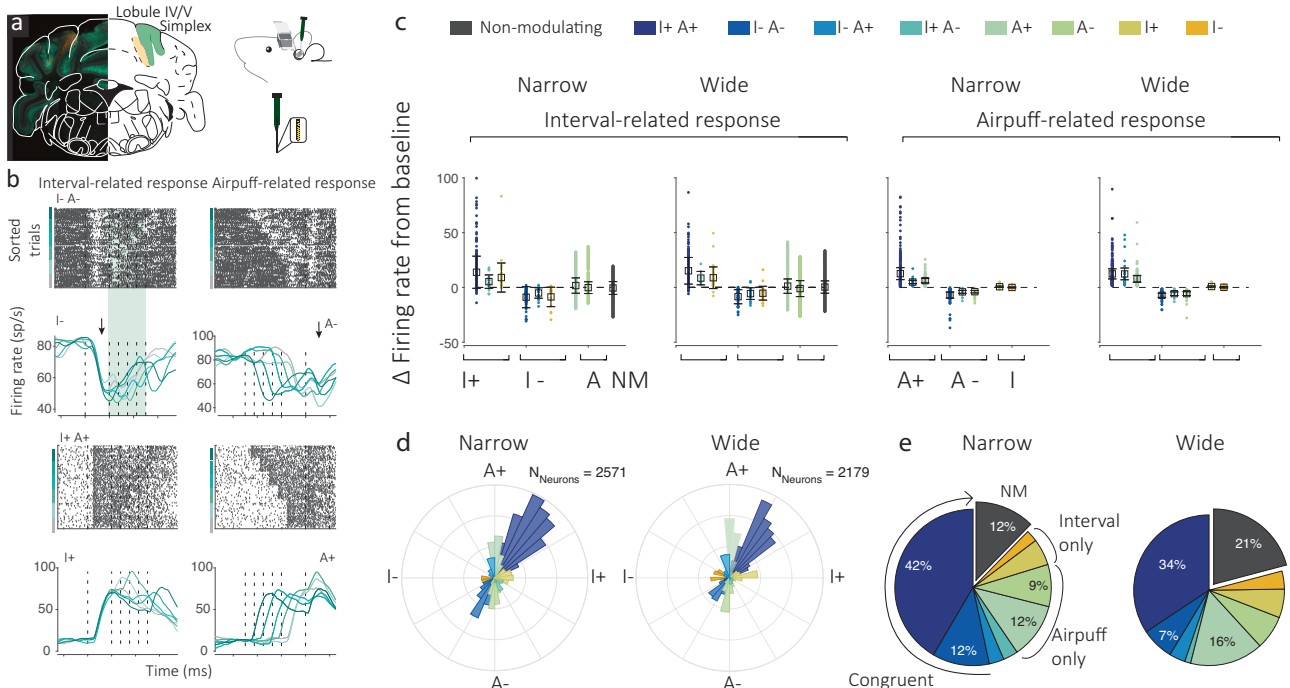

**Fig. 2 | Classification of neural responses. a** Recordings were made with a grid-in-chamber setup in head-fixed animals. Probe entry locations as marked with DiI at grid point epicenter on the last day of recording. **b** Classification of neurons based on prior-related and airpuff-related responses. PSTHs or rasters are aligned to either epoch. A classification technique that quantified firing rates in each epoch with respect to baseline firing rate determined whether the neuron significantly responds to either epoch and in which direction. I indicates interval or ISI-related activity, A indicates Airpuff-related activity and + and − signs indicate facilitation and suppression, respectively. **c** Relative firing rate of each functional class with respect to baseline. Dots represent individual neurons (total Nnarrow = 2571,

Nwide = 2179). Boxes represent averages. Error bars indicate standard error of mean. Dashed line indicates no change from baseline. NM stands for non-modulating. **d** Polar histogram exhibiting functional heterogeneity in recorded cerebellar cortical neurons for the Narrow (left) and Wide (right) condition. Classification of Purkinje cells based on their response to the ISI epoch (I+ or I−) and Airpuff epoch (A+ or A−). **e** Relative proportion of each functional class among recorded neurons for the Narrow (left) and Wide (right) conditions. Congruent conditions (I+A+ and I-A−), Interval only (I+ I−), and Airpuff only (A+ A−) are marked. Incongruent conditions, I+A- and I-A+ (light blue colors) are not labeled. Colors indicate functional classes in the same manner as the legend in **c**.

Wide (5%) conditions (Fig. 2e). A relatively large proportion of neurons modulated only for the Interval epoch in the Narrow (8%) and Wide (10%) conditions and also many neurons modulated only in response to the Airpuff epoch in the Narrow (21%) and Wide (23%) conditions. Only 12% of neurons in the Narrow and 21% of neurons in the Wide condition were found to be non-modulatory in either epoch (Fig. 2e).

**Cerebellar motif models explain heterogeneous functional representations**

Previous work in eyeblink conditioning has only reported facilitating and suppressing Purkinje cells underlying predictive eyelid behaviors[23,37,38] and characterization of molecular layer interneuron physiology has been limited during such behaviors. Here, we report eight functional classes of modulation in response to associative learning of well-timed eyelid responses with different temporal statistics. To explain the observed heterogeneity, we leverage insights from recent functional anatomical discoveries of collateral connectivity among Purkinje cells and MLIs[6,8,39] to develop a motif microcircuit model, called TRACENet, that accounts for different types of local feedforward inhibitory, disinhibitory, and recurrent motifs.

Purkinje cells receive inputs from granule cell parallel fibers and from climbing fibers, which influence the generation of high-frequency simple spikes and low-frequency complex spikes, respectively. Complex spike-specific tuning has been shown to be important in understanding behaviors such as motor adaptation[26], motor learning[40,41], and recently, non-motor behaviors[42,43]. Based on previous reports[23,41] and observations in our own data, the model assumes that the airpuff elicits a complex spike in the Purkinje cell. It is also assumed that Purkinje cells receive a cascade of parallel fiber signals following the light cue,

which is consistent with recent evidence of activity patterns in the granule layer during different behaviors in various species[44–46]. Conjunctive activation of both these pathways is believed to lead to long-term depression of the postsynaptic terminals of PF-PC synapses for which parallel fibers were active within a brief eligibility window. In the model, this leads to well-timed suppression of Purkinje cell activity, which has been hypothesized as the primary driver for conditioned eyelid responses. Many of these aspects of the model are similar to previous proposals in the field[47–50]. However, while suppression of Purkinje cell activity is believed to be the long-standing model for learning in such behaviors, there is no explanation for the existence and role of the many facilitating Purkinje cells that are also found during such behaviors[23,38]. Further unlike previous models, TRACENet takes relatively recent reports of diversity in mossy fiber inputs[51] and climbing fiber inputs to both the Light and Airpuff cues[23,41] into account for generating its neural activity predictions (Fig. 4b and Supplementary fig. 6).

Furthermore, TRACENet advances computations from a single Purkinje cell to its local microcircuit, i.e. it describes how recurrent and feedforward connectivity with other Purkinje cells and MLIs may propagate a learned signal further and possibly account for the rich and heterogeneous dynamics that we observe. In addition to local connectivity, we assume that within a lobule, not all Purkinje cells will receive a task-related climbing fiber signal. We also assume that not all molecular layer interneurons and Purkinje cells in a local population will receive relevant airpuff or light-related signals. Based on these assumptions, the model constructs feedforward inhibitory and low-gain recurrent motifs (Fig. 4a, b, see "Methods" section). These entail feedforward PC-MLI and recurrent PC-PC connectivity. These

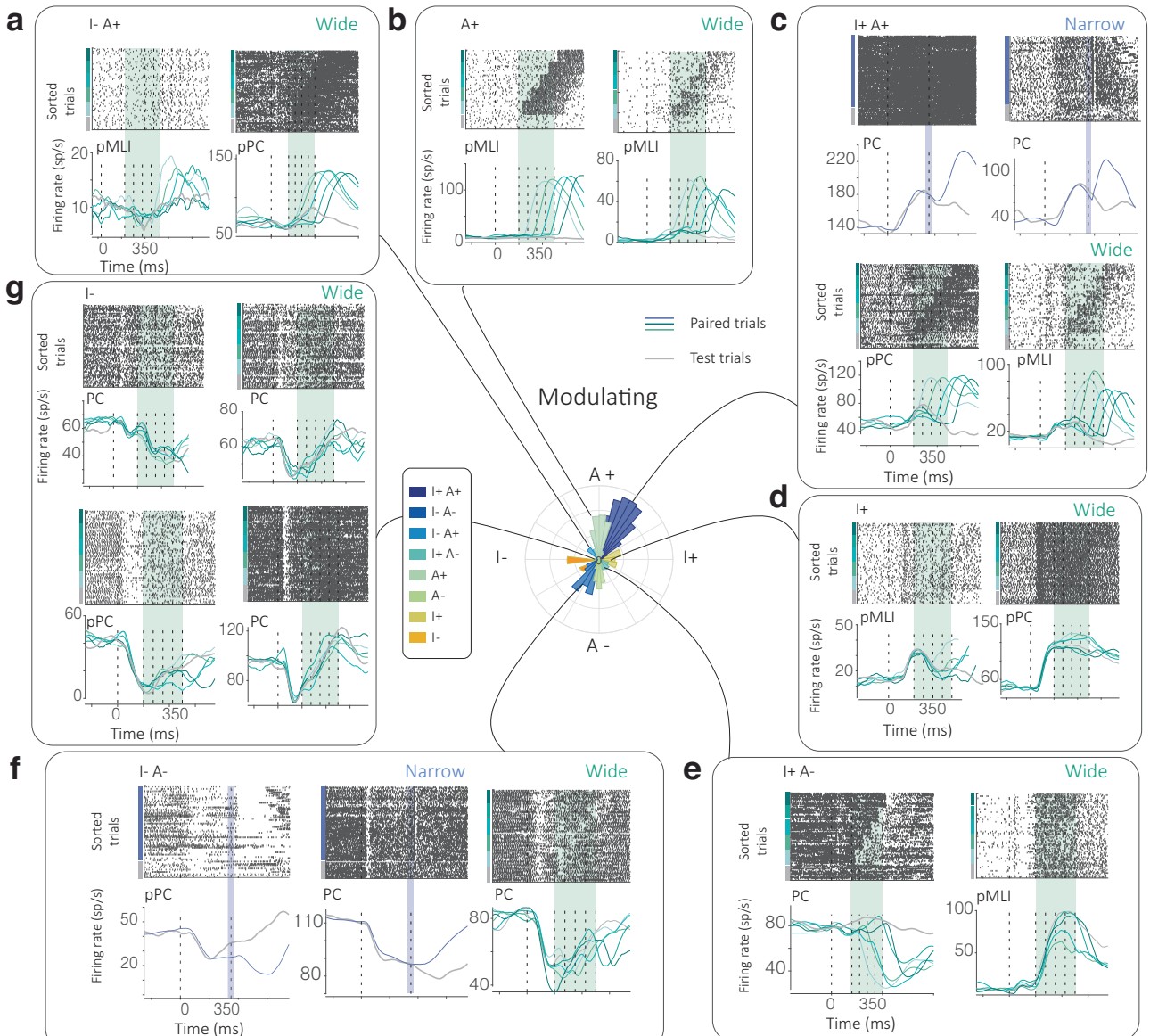

**Fig. 3 | Neural activity profiles for various functional classes. a–g** A polar histogram, indicating the various functional classes identified in an individual, is presented at the center with dark blue-orange colors indicating functional classes. For these, rasters and firing rates for neurons are provided in clockwise order, for class I−A+, A+, I+A+ (for both the Narrow (blue) and Wide condition (green)), I+, I+A −, I–A− and I−. Gray lines indicate test trials whereas blue/green lines indicate firing rates on paired trials for each condition. Activity is aligned to the onset of the Light cue (0 ms, dashed line) and the mean of both conditions is marked at 350 ms. Times of airpuff(s) for each condition are also indicated as dashed lines.

connectivities and assumptions are sufficient to account for all eight functional classes observed in the neural data for both the Narrow and Wide conditions (Fig. 4c, d and Supplementary fig. 6).

### Locus of Purkinje cell learning of temporal stimuli examined by chronic silencing

While delay eyeblink conditioning has been well-studied in the cerebellar cortex[23,37], we know relatively little about the cerebellar processes that support trace conditioning[52]. Recent work has shown that the neural mechanisms for trace and delay conditioning are expected to be similar in cerebellar circuitry[53]. However, it remains unclear whether trace conditioning engages larger proportions of rodent cerebellar lobules than previously studied in delay conditioning behaviors, in which animals are trained for considerably shorter durations.

To examine the anatomical locus associated with trace eyeblink conditioning behavior, we used targeted chronic lesioning of

Purkinje cells in lobules IV/V and Simplex (Fig. 5a–c and Supplementary fig. 7a–d). We induced cell death selectively in Purkinje cells throughout lobules Simplex and parts of lobule IV/V by introducing Diptheria Toxin A (dTA) in them via retrograde transport from the anterior interposed nucleus. This was enabled using a PCP2-Cre mouse line, which expresses Cre in Purkinje cells, in combination with a viral construct that facilitates Cre-dependent expression of dTA. In a control group, we lesioned Purkinje cells projecting to the posterior interposed nucleus (Fig. 5b and Supplementary fig. 7b, d). Using a cell death marker (GFAP[54]) as a positive control, missing fluorescent protein (GFP) expressing only in Purkinje cells as a negative control, presence of viral particles (mCherry) as a second positive control, we were able to quantify and detect the location of ablated Purkinje cells throughout lobules Simplex and IV/V (Fig. 5c and Supplementary fig. 7a–d) for the test group. We performed a similar analysis for the posterior cerebellum for the control group.

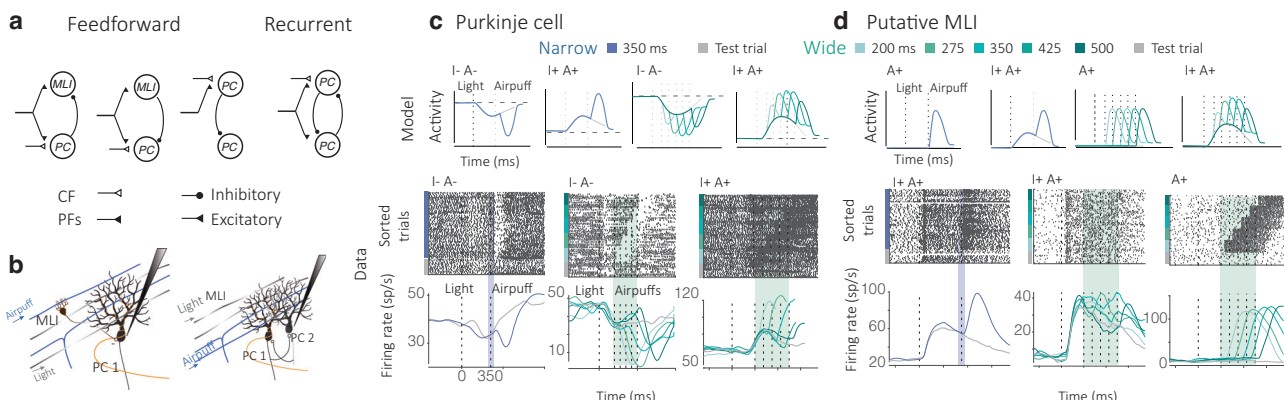

**Fig. 4 | Local microcircuit motif computations explain heterogeneity of cerebellar cortical responses. a** Motif representations of feedforward and recurrent connectivity between Purkinje cells (PCs) and molecular layer interneurons (MLIs) alongside examples of parallel fiber (PF, filled black triangle) and climbing fiber (CF, unfilled black triangle) inputs. **b** Illustrations of local cerebellar network anatomy with feedforward and recurrent connections between Purkinje cells (PC) and molecular layer interneurons (MLIs). **c** TRACENet predictions for Purkinje cells connected in feedforward and recurrent patterns with MLIs and PCs for the Narrow (blue) and Wide (green) conditions. Below: Purkinje cell rasters and firing rates are shown for the Narrow (blue) and Wide (green) conditions for paired (blue/green) and test trials (gray). **d** TRACENet predictions for MLIs connected in feedforward patterns with PCs for the Narrow (blue) and Wide (green) conditions. Below: Putative MLI rasters and firing rates are shown for the Narrow (blue) and Wide (green) conditions for paired (blue/green) and test trials (gray).

Next, we examined the effect of these lesions on predictive eyelid behavior in the test and control groups (Fig. 4d, e and Supplementary fig. 8a–f). We found a significant decline in the performance of the test group after the viral injection (early and late post-injection phases, see methods), registering a significant main effect for the percentage of predictive eyeblinks ($F(2,6) = 4.89$, $p = 0.04$) and a significant decline in their amplitude ($F(2,6) = 4.75$, $p = 0.047$; Fig. 5d, e) for the test group and no significant main effects for the control group, which affected hemispheric lobules in the posterior cerebellum (Supplementary fig. 7a–d). Since these locations coincide with our recording sites, we inferred from this that Purkinje cells in lobule IV/V and Simplex play a role in supporting predictive eyelid behaviors in our modified trace conditioning task.

## Discussion

Previous work in cerebellar physiology has reported cerebellar cortical neurons as facilitating or suppressing their dynamics during a variety of behaviors[13,15,17,23–25,27,37], which, in the case of Purkinje cells, are also referred to as *burst* and *pause* firing patterns. Recent discussions in the field suggest that more complex physiological patterns ought to be expected based on recent revelations of feedforward and feedback motif connectivity among Purkinje cells and MLIs[6,8,39]. Here, we report eight functional representations of cerebellar cortical activity underlying a relatively simple associative behavior, suggesting that both Purkinje cell and molecular layer interneuron dynamics may be more complex and heterogeneous than previously believed.

We use recent work to develop local motif models that can account for such heterogeneity. Our modeling suggests that a subset of Purkinje cells could perform an initial learning function due to long-term plasticity induced by conjunctive activation of climbing fiber and parallel fiber pathways. This learning, however, then propagates to neighboring neurons via feedforward connectivity to MLIs and potential recurrent connectivity to Purkinje cells. Our motifs, however, do not capture many nuances and findings observed in recent work[6,8,39], for example, multisynaptic interactions, some of which remain unreconciled. This may explain why there are some forms of heterogeneity observed in our data that our models and classification methods are, at present, unable to account for, for e.g., diversity in temporal dynamics of the activity during each epoch or complex nonlinear activity patterns within each epoch.

Recent advances in neurophysiology techniques are enabling finer scrutiny of the function of local microcircuits just as our ability to record from large populations of neurons is increasing in capacity, leading to a more complex picture of neural structures and their function. At the same time, computational modeling serves to demystify how seemingly complex neural circuits could amplify, sieve, and refine relevant outputs to their downstream targets. Here, we provide evidence of functional heterogeneity in cerebellar cortical circuits during temporal behaviors and suggest that similar diversity may be found in a variety of other behaviors.

## Methods

### Mice and surgical setup

This study utilized fourteen mice (Postnatal age >60 days, 5 female). Six C57BL/6 mice were used for the behavioral training sessions and were used for physiological recordings in lobules IV/V and Simplex of the cerebellar cortex. Eight PCP2-Cre-eYFP mice were used for test and control conditions of viral lesioning experiments using diphtheria toxin A ($N = 8$). All procedures were performed in accordance with protocols approved by the animal care and use committees (IvD) at Erasmus Medical Center. Mice were housed in a 12:12 light: dark cycle and were tested in the light phase. No restriction was placed on food. Water dispensation was maintained throughout and body weight was monitored. All surgical procedures were carried out aseptically under a mixture of 3% isoflurane in 1l/min oxygen anesthesia. Post-operative analgesia management was enabled by administering Buprenorphine HCl (0.1 mg/kg) and Carprofen (5 mg/kg). Mice were monitored for 3 days post-surgery and were provided analgesic medication (Rimadyl, Carprofen, Zoetis B.V) before the continuation of experiments. Between postnatal ages P60 and P80, a semi-magnetized pedestal was installed on the skull to enable head fixation during behavioral training. A custom-designed thermoplastic recording chamber was installed on a cerebellar craniotomy centered at AP −6.25 mm, ML −2.25 mm, and 3 mm in diameter. The chamber was affixed using an adhesive (Optibond, Kerr corporation, USA) and dental cement (Charisma, Kulzer, USA). After implantation, the chamber of each mouse was cleaned daily with saline and a dura-cleaning tool and disinfected with low concentrations of ethanol to maintain the hygiene of the dura and surroundings. All surgical installations on the skull were aligned using 2D line level. During recording sessions, a custom-made plexiglass grid was installed into the chamber to ensure systematic anatomical access to cerebellar structures and to provide stable housing for the electrode.

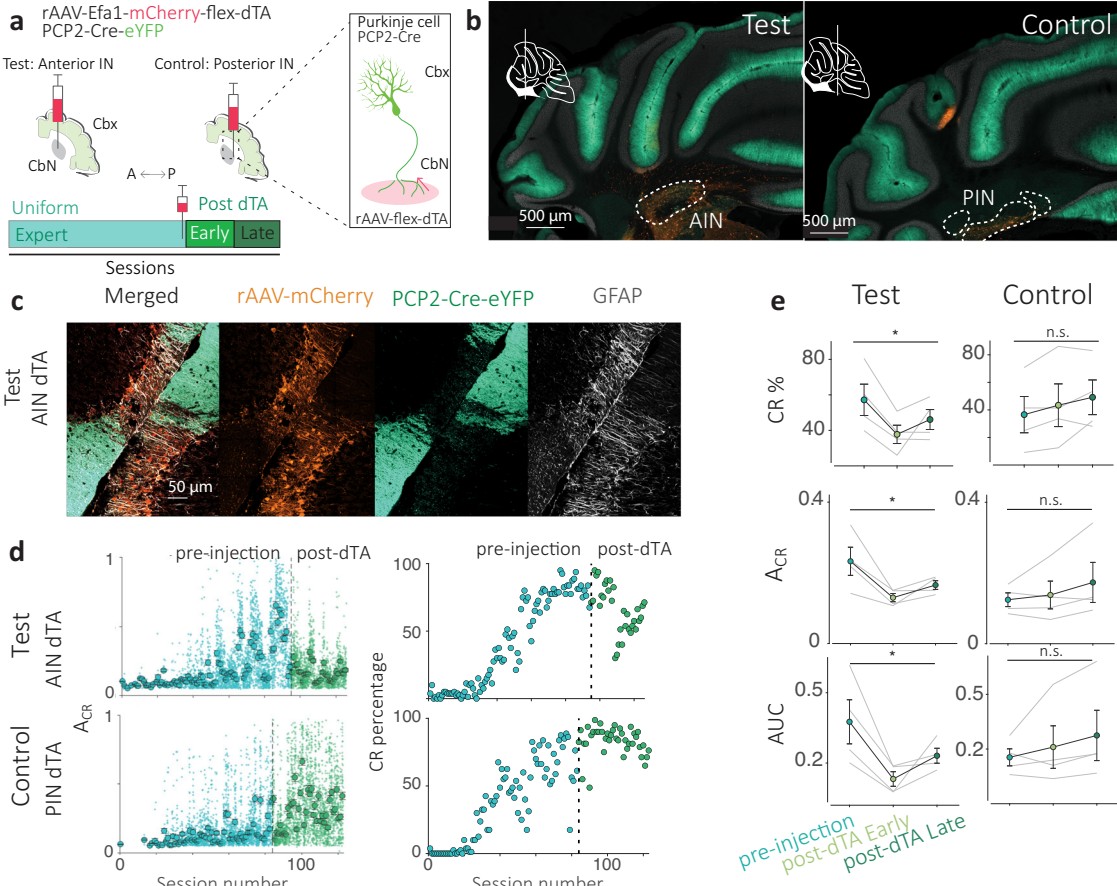

**Fig. 5 | Chronic perturbation of Purkinje cells in cerebellar lobule simplex.**
**a** Targeted chronic lesions of Purkinje cell populations were enabled by a retrograde viral vector encoding a Cre-dependent construct for Diptheria Toxin A (dTA) and mCherry (rAAV-mCherry-flex-dTA). Mice trained on the Wide condition (PCP2-Cre-eYFP) were injected after the crystallization of behavioral performance metrics. Progression is denoted in three stages: pre-injection (blue), early post (light green, began 7–10 days after injection), and late post-injection (dark green). **b** A coronal section from the test group (left) shows that the construct was injected (in orange, rAAV-mCherry) in the anterior interposed (AIN) the downstream target of lobule simplex (inset marks A-P position on a schematic of a sagittal section) Purkinje cells (green, indicated Cre and eYFP). In contrast, a coronal section from the control group (right, $N = 4$) with the same staining shows that the construct was injected in the posterior interposed (PIN). **c** A confocal image of the Simplex lobule for the test group shows that the presence of mCherry (orange), indicated viral particles, coincides with the absence of Purkinje cells (green), and the presence of a cell death marker glial fibrillary acidic protein (GFAP in white). Results were confirmed in $N = 4$ mice. **d** Behavioral metric comparison for $A_{CR}$ and predictive response percentage (CR percentage) within training sessions. Large circles indicate session averages and dots indicate individual trials in the pre (blue) and post (green) injection phases. **e** Metric comparisons for test (left) and control (right) groups for pre, early and late post-injection stages for $A_{CR}$ (ANOVA $F(2,6) = 4.75$, $p = 0.047$) and CR percentage (ANOVA $F(2,6) = 4.89$, $p = 0.04$). '*' represents an alpha of 0.05. Circles connected by black lines indicate averages and gray lines indicate individuals. Error bars represent standard error.

## Task, experimental setup, design, and training

Mice were head-fixed to a post with a semi-magnetized pedestal attached to their skull and were able to comfortably rest or move freely on a self-initiating treadmill with low forward resistance but adequate textured grip. A high-speed infra-red camera (Basler Ace aca1300, Basler, Germany) was trained on the eye of the animal to record movements. Our setup was designed based on earlier proposals for similar tasks[55]. Posterior whiskers were trimmed to minimize interference with eyeblink detection. A custom-made device delivered a periocular airpuff representing the unconditioned stimulus using an air-pressurized drive triggered by a 5V pulse. A semicircular array of white LEDs was used to deliver the light representing the conditioned stimulus. This delivery system was built to ensure homogenous and adequate bilateral visual input to the mouse (to ensure equivalent bilateral activation of the pontine nuclei). The camera recording and stimulus delivery system were integrated using custom drivers and code in Objective C (Cocoa framework XCode, Apple, Cupertino, USA) and Matlab 2020a (Matlab, Nattick, USA). Pulse pal (Sanworks, USA) was used for regulating stimulus delivery.

Each experimental training session lasted for 80 trials, with an inter-trial interval sampled from a discretized truncated exponential function (tau = 4s). Mice were considered trained when the CR percentage values saturated to or exceeded a threshold of approximately 40% (Supplementary Fig. 2). After performance metrics stabilized, mice underwent a surgical craniotomy and chamber placement.

On each trial, the LEDs (Light) were active for 70 ms, followed by the interstimulus interval (ISI), after which the airpuff was administered for 70 ms, resulting in reflexive eye closure. The ISI was determined based on the time interval distribution condition. For the narrow condition, the ISI was 350 ms, whereas in the wide distribution, it was sampled with uniform probability from a discrete distribution (200, 275, 350, 425, and 500 ms), which is shown to maximize training efficiency[56]. At the start of each session, two airpuff-only trials were administered to compute and calibrate the baseline eye closure by calculating the change in pixel value during each frame in the region of interest. This value was used to normalize subsequent eyeblink responses. Mice were monitored at all times during training and were given a time-out if squinting or extended eye closure was detected.

Sessions also consisted of paired and test trials, where the airpuff was omitted.

## Quantification of eyeblink metrics

Eyeblink responses within the session were normalized to the reflexive response evoked by the airpuff either in absence of the light stimulus on airpuff-only trials or on paired trials. A predictive response was detected if the eyeblink trace velocity increased beyond a threshold and amplitude that exceeded thrice the baseline standard deviation before the light presentation across all trials within a session. Percentage of the predictive response within a session was computed as the number of trials where such a response was detected against the total number of trials within a session. Response amplitude was computed as the maximum eye closure of the predictive component after light onset.

## Large-scale electrophysiology using silicon probes

Extracellular recordings were performed using ESSY-37 E1 32 channel silicon probes (Cambridge NeuroTech, UK). An Intan (RHD2132, USA) amplifier was used to digitize and amplify the recorded extracellular voltage signals at 16 bit and were recorded using an Intan RHD2000 Amplifier Evaluation System (sampling rate: 30,000 Hz). We used Open Ephys[57] for acquisition, online monitoring, and processing of cerebellar electrophysiological signals. A craniotomy 3 mm in diameter was made at AP −6.25 mm, ML −2.25, after which a cylindrical light-weight recording chamber with a sealable lid was installed at the rim of the craniotomy on the skull surface. During recordings, lidocaine was applied onto the dura surface 15 min before recordings, which was subsequently cleaned with saline. A plexiglass grid, designed to fit into the chamber at a horizontal orientation was assembled and the silicon probes were lowered through the grid holes. From previous anatomical mappings, we determined that the grid access points coincided with lobule IV/V and Simplex. Daily penetrations followed lateral to medial increments within and across gridpoints 4 and 5. On a given day, 1–4 sessions were recorded at different depths unidirectionally from ventral to dorsal. The silicon probe was allowed to stabilize for 20 min before recording. The mouse could move freely on the treadmill at all times without influencing probe stability, however, during trials, the wheel movement was arrested.

## Unit isolation, Purkinje cell identification, and putative molecular layer interneurons

In the absence of optogenetic manipulations, Purkinje cells can also be identified from large-scale in vivo recordings through physiological metrics. We used five criteria for such identification: (1) Recordings were performed from the molecular layer and Purkinje layer based on the polarities of the identified complex spike patterns. (2) The baseline firing rate of neurons lay between 40-200 Hz. (3) Complex spikes and simple spike waveforms were recorded from the same channel or adjacent locations within 20 μm. (4) The complex and simple spike waveforms conformed to standard timescale and shape properties[43]. (5) The complex spike elicited a 15-20 ms simple spike suppression in a cross-correlogram (with a 10% contamination rate for this criterion, Supplementary fig. 7). Purkinje cells for which the last condition was not met were labeled putative Purkinje cells. Note that we were unable to record granule cells due to limitations in electrode impedance and data from recording locations in granule layers were excluded. Although we only record from the molecular layer, neurons that we could not classify as Purkinje cells are referred to as putative molecular layer interneurons.

## Functional classification of cerebellar cortical population

Modulatory activity in each subclass was determined in two ways, 1) by computing variance across binned trials (15–20 trials) within conditions and by computing their deviation in the category-specified

directions. 2) By fitting a GLM model to the neural data for the Airpuff, Interval epochs (similar to previous work[36,58]). The results were validated by plotting the full distribution of firing rates and weights for each component in each category (Supplementary fig. 5a, b). This resulted in 8 functional types, depending on valence/weight of the interval or airpuff-related epochs. We also characterize one generic non-modulatory class in addition to the eight modulatory types.

## Injections and immunohistochemistry

We injected rAAV- EF1a-mCherry-flex-dTA into the anterior or posterior interposed of PCP2-cre-eYFP mice for the test and control groups, respectively. We injected 50–80 nl of the viral construct using a microinjection device (Nanoject II, Drummond scientific, USA) using 20 μm glass tip pipettes at a depth of 2250 μm μm for the anterior interposed (AP −6.20 mm, ML −2.25,) and 2300 μm for the posterior interposed (AP -6.35 mm, ML -2.25). Each volume was administered over 3–4 injections at a speed of 23nl/s with >10 min between each injection and before gradual withdrawal. Three days post craniotomy, mice resumed the same training protocol on the Wide condition. In the early-late post-injection analysis, we did not include 7–10 days of post-injection behavioral data owing to the time that the AAV vectors need for effective expression. After around 20 days post-injection, mice were deeply anesthetized with an overdose of pentobarbital (0.2 ml, i.p.) and transcardially perfused with 20 ml saline followed by 50 ml 4% paraformaldehyde (PFA) in PBS. Brains were extracted and post-fixed in 4% PFA for 2h and incubated in 10% sucrose overnight at 4 °C. Brains were then embedded in gelatine and cryoprotected in 30% sucrose in PB, frozen on dry ice, and sectioned using a freezing microtome (50 μm thick). For immunohistochemistry in the dTA experiments, sections were blocked for 1h at room temperature in PBS with 0.4% Triton X-100 and 10% N-hydroxysuccinimide (NHS) solution and incubated for 48 h at 4 °C in a primary antibody (rabbit anti-GFAP antibody, 1:7000, Agilent Technologies Inc.) diluted in PBS with 2% NHS and 0.4% Triton X-100. These sections were then washed and incubated for 2h at room temperature in the secondary antibody (donkey-anti-rabbit, 1:400, Jackson). In all experiments, slices were counterstained with DAPI (1:100.000, Invitrogen) and mounted using Mowiol (Sigma). All sections were imaged using a fluorescent microscope (Axio Imager 2, Zeiss) operated by ZEN 2.6 Pro (Zeiss). All sections were then registered to the Allen Common Coordinate Framework CCF using the AllenCCF software[59]. Probe tracks and histological labels were quantified using this tool and further analyzed for location analysis.

## TRACENet model

**Encoding model for time intervals and scalar variability.** We assume granule cell spike counts (r) obey an inhomogeneous Poisson process whose rate function is Gaussian with mean $\omega(t)$ and standard deviation $\sigma_i$. The maximum firing rate of the $i^{th}$ granule cell parallel fiber, $\mu_i$, is specified with respect to the onset of the light or conditioned stimulus.

$$p(r|t) = \prod \frac{1}{r_i!} \omega_i(t)^{r_i} e^{-\omega_i(t)} \tag{1}$$

$$\omega_i(t) = \frac{1}{\sqrt{2\pi\sigma_i^2}} e^{\frac{-(t-\mu_i)^2}{2\sigma_i^2}} \tag{2}$$

Due to scalar variability, the internal estimate of elapsed time ($\tilde{t}$) has a probabilistic relationship to the chronometric elapsed time ($t$). We formulated this relationship as a conditional Gaussian probability distribution whose mean is $t$ and whose standard deviation remains constant for the $i^{th}$ kernel but scales across kernels by linear scaling factor $w_b$, equivalent to the Weber fraction that best describes behavioral observations. Therefore, we assume a heterogeneous

population that takes such a form and approximates Weber's law.

$$p(\tilde{t}|t) = \frac{1}{\sqrt{2\pi(w_b t_i)^2}} e^{\frac{-(\tilde{t}-t_i)^2}{2(w_b t_i)^2}} \qquad (3)$$

We will now assume a relatively dense and discrete heterogeneous population over stimulus time $t_s$. Let $p(t_s)$ be the prior probability of the stimulus time $t_s$. While each granule cell may have a preferred firing time, only a subset of granule cells will be active (over elapsed time) when a given $t_s$ is administered.

The firing rate reduction over the population was modeled by a gain function, $g(t)$, with time constant $\tau_{basis_i}$, and the increase in width was modeled linearly before learning, $\sigma_{basis_i} = \sigma_0 i\kappa/N$, where $i$ indexes neurons ordered according to their preferred time interval, $N$ is the total number of neurons and $\kappa$ is the proportion of increase in the width $\sigma_0$. The resulting function describes the rate of the $i^{\text{th}}$ granule cell (GC).

$$r_i(t) = g(t) \frac{1}{\sqrt{2\pi\sigma_{basis_i}^2}} e^{-(t-t_s)^2/2\sigma_{basis_i}^2} \qquad (4)$$

For encoding of $p(t_s)$, we define $w_i$, which represents the postsynaptic weight of the $i^{\text{th}}$ GC with a Purkinje cell. Long-term depression (LTD) in TRACE is modeled for each GC-PC synapse as proportional to the rate of firing of respective GCs shortly before the firing of climbing fibers (CFs) at the time of the airpuff. The time before CF firing at which GC-PC synapses become eligible for LTD is called the eligibility trace ($\epsilon$), which we assume occurs 20-50 ms before the onset of the airpuff in the model. In the absence of CF stimulation and in the presence of GC firing, Long Term Potentiation (LTP) is induced. The dynamics of LTD and LTP were governed by their respective time constants, $\tau_{ltd}$ and $\tau_{ltp}$. In the absence of learning, synapses would gradually drift toward the baseline, $w_0$.

$$\frac{dw_i}{dt} = -\frac{1}{\tau_{ltd}} r_i(t_s - \epsilon)\delta(t - t_s) + \frac{1}{\tau_{ltp}}(w_0 - w_i) \qquad (5)$$

At steady state, the sum of all weights over the basis set population resembles the shape of an inverted prior distribution p(t), which makes sense given that Purkinje cells are inhibitory and one of the primary learning mechanisms in the cerebellar cortex is long-term depression[60] of Purkinje cell activity. In the model, the change in baseline PC activity is computed as a weighted sum of GC activity.

$$\Delta V_{pc}(t) = \sum_{i=1}^{N} r_i(t)w_i \qquad (6)$$

Feedforward PC-PC motifs: We model the disinhibitory influence of PC1 on a neighboring PC2 by subtracting the activity of PC1 from the baseline activity of PC2 and using a feedforward gain ($\kappa_p$), which represents the strength of the synaptic connectivity of PC1 on PC2:

$$V_{pc2}(t) = \kappa_p(V_{pc2}(0) - V_{pc1}(t)) \qquad (7)$$

Feedforward PC-MLI motifs: We model the disinhibitory influence of PC1 on a neighboring MLI by subtracting the activity of PC1 from the baseline activity of the MLI and using a feedforward gain ($\kappa_m$), which represents the strength of the synaptic connectivity of PC1 on the neighboring MLI:

$$V_{MLI}(t) = \kappa_m(V_{MLI}(0) - V_{pc1}(t)) \qquad (8)$$

The reciprocal connectivity between MLI-PC can also be modeled using the same principle.

Recurrent PC-PC motifs: Utilizing insights from previous work[61], we model a mutually inhibitory interaction between two Purkinje cells

(v and u), which receive common parallel fiber input ($\theta$). Each neuron has its own time constant ($\tau$) and input ($w_\theta$) and recurrent weights($w_{uv}$), which we assume to be symmetric. The time-varying firing rates of these units are then governed by:

$$\tau_{PC1}\dot{u} = -u + f(w_{u\theta}\theta - w_{uv}v) \qquad (9)$$

$$\tau_{PC2}\dot{v} = -v + g(w_{v\theta}\theta - w_{uv}u) \qquad (10)$$

where,

$$f(x), g(x) = \frac{1}{1+e^{-x}} \qquad (11)$$

## Reporting summary

Further information on research design is available in the Nature Portfolio Reporting Summary linked to this article.

## Data availability

The data generated in this study have been deposited in the Dryad database (https://doi.org/10.5061/dryad.41ns1rnmc).

## Code availability

Code is provided on https://github.com/NarainNeuro/CerebellarHeterogeneity_NC_2023.

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

## Acknowledgements

The authors thank Gerard Borst for his comments on the manuscript. We also thank Dick Jaarsma for reagents and Stephanie Dijkhuizen for help with behavioral training. This work was supported by the ERC Advanced, Proof of Concept grants (EU), ZonMW (NWO), Intense (NWO), BIG (Erasmus MC), 3V-Fonds (KNAW), the Medical Neuro-Delta (EU), and Gravitation (NWO DBI[2]) grants to C.D.Z. and the MSCA reintegration grant (EU PredOpt 796577), Vidi (NWO VI.193.076), Aspasia (NWO 015.016.012), and Gravitation (NWO Dutch Brain Interface Initiative DBI[2] 024.005.022) grants to D.N.

## Author contributions

D.N. and C.D.Z. conceived the project. D.N. built the setup, performed the electrophysiological experiments, and built the computational models. J.K. and D.N. analyzed the data with help from G.G.B. and M.R.; M.R. and G.G.B. performed all histological and microscopy analyses. D.N. supervised J.K., G.G.B., and M.R. and provided financial support for the project. D.N. wrote the manuscript with input from all authors.

## Competing interests

The authors declare no competing interests.
