## [Peer Review File · Nature Communications]

Heterogeneous encoding of temporal stimuli in the cerebellar cortexEditorial Note: This manuscript has been previously reviewed at another journal that is not operating a transparent peer review scheme. This document only contains reviewer comments and rebuttal letters for versions considered at *Nature Communications*.

REVIEWERS' COMMENTS

Reviewer #2 (Remarks to the Author):

The manuscript "Heterogeneous encoding of temporal stimuli in the cerebellar cortex" by Narain and colleagues presents an important conceptual advance in considering how Purkinje neurons encode information to support associative learning. By considering recurrent circuit motifs they find a mechanism that could explain the diverse patterns of Purkinje neuronal activity observed in vivo. The manuscript is thorough and rigorous. The data are clear and convincing. I think this study will be impactful in the field and inspire future studies. The revisions fully address any concerns.

Minor:

In the main text it would be useful to clarify a few more details of the lesion experiments, including that the experiments utilize a cell-type specific toxin under the control of PCP2-cre that is targeted to Purkinje neurons relevant to the task through retrograde infection. These details help a lot in following the experiment but are only interpretable after reading the methods.

Reviewer #3 (Remarks to the Author):

This manuscript describes experiments that involved recordings from Purkinje cells (PCs) during expression of conditioned eyeblink response. A simple behavioral regimen was employed to create eyeblink conditioned responses (CRs) with different temporal properties: a fixed temporal relationship between light conditioned stimulus and air puff unconditioned stimulus was used to produce "narrow" CRs, and a variable relationship was used to produce "wide" responses. The major finding is that the PCs displayed a heterogeneous set of responses under these conditions, with some increasing their activity, others decreasing, etc. The main conceptual claims are that we need to retool the way we think about the cerebellum given this heterogeneity and that their computational model of portions of the cerebellar circuitry predict this heterogeneity of responding.

With apologies to the authors, I find these claims rather a stretch and an overly dressed-up presentation of things already known.

First, the observation that PCs displayed heterogenous responses: of course they did. It's well established, from work using several different cerebellar-dependent behaviors, that the responses of PCs are related to the circumstances that activate their climbing fiber input. This has been shown clearly and quantitatively for VOR adaptation, for saccade adaptation and even for eyeblink conditioning. Absent this identification we don't know what output or response the PCs control, and thus we might expect a wide variety of responses with most synchronized to inputs (such as lights and air puffs) rather than to responses.

Second, the computational model the authors present represents in a primitive way processing within a parasagittal stripe. Thus, their "predictions" are only relevant to heterogeneity of responding from PCs within the same stripe, and thus those whose climbing fiber inputs respond to similar stimuli. The model is dressed up with a bit of math, but it is a fairly primitive expression of ideas and computational models that have already been published.

- 1) It assume granule cells fire at different times during a CS
- 2) PCs use LTD/P to learn responses from eyelid conditioning training
- 3) PC-PC and PC-MLI interactions provide ways for PCs to have different responses, which is rather obvious given that in the model PCs can inhibit other PCs.

In sum, because previous work has shown homogeneity of responding within a stripe during eyeblink and other tasks, and given that the authors have not sorted or categorized their PCs according to the climbing fiber responses, the heterogeneity appears to be the result of recording from many PCs that have various and different jobs to do. The computational model is designed to explain heterogeneity within a parasagittal stripe, which is contradicted by previous work and is not demonstrated here. In addition, the primitive nature of the model quite belies the idea that it can have much predictive value anyway.

Reviewer #2 (Remarks to the Author):

The manuscript "Heterogeneous encoding of temporal stimuli in the cerebellar cortex" by Narain and colleagues presents an important conceptual advance in considering how Purkinje neurons encode information to support associative learning. By considering recurrent circuit motifs they find a mechanism that could explain the diverse patterns of Purkinje neuronal activity observed in vivo. The manuscript is thorough and rigorous. The data are clear and convincing. I think this study will be impactful in the field and inspire future studies. The revisions fully address any concerns.

Minor:

In the main text it would be useful to clarify a few more details of the lesion experiments, including that the experiments utilize a cell-type specific toxin under the control of PCP2-cre that is targeted to Purkinje neurons relevant to the task through retrograde infection. These details help a lot in following the experiment but are only interpretable after reading the methods.

We thank the reviewer for their comments and have implemented their suggestion about adding details about the lesion experiments in the text.

Changes made:

"We induced cell death selectively in Purkinje cells throughout lobules Simplex and parts of lobule IV/V by introducing Diphtheria Toxin A (dTxA) in them via retrograde transport from the anterior interposed nucleus. This was enabled using a PCP2-Cre mouse line, which expresses Cre in Purkinje cells, in combination with a viral construct that facilitates Cre-dependent expression of dTxA."

Reviewer #3 (Remarks to the Author):

This manuscript describes experiments that involved recordings from Purkinje cells (PCs) during expression of conditioned eyeblink response. A simple behavioral regimen was employed to create eyeblink conditioned responses (CRs) with different temporal properties: a fixed temporal relationship between light conditioned stimulus and air puff unconditioned stimulus was used to produce "narrow" CRs, and a variable relationship was used to produce "wide" responses. The major finding is that the PCs displayed a heterogeneous set of responses under these conditions, with some increasing their activity, others decreasing, etc. The main conceptual claims are that we need to retool the way we think about the cerebellum given this heterogeneity and that their computational model of portions of the cerebellar circuitry predict this heterogeneity of responding.

Our view of the finding here is that PCs do not merely increase or decrease their activity during various forms of eyeblink conditioning, which was established previously by Heiney et al 2014, Halverson et al 2015, and in work from our own lab (Ten Brinke et al 2015). Here we show that cerebellar cortical neurons, PCs and putative MLIs, show mixed and multiplexed task representations with significant functional diversity i.e. a neuron can congruently facilitate in both the interval and airpuff epochs or

incongruently facilitate its activity in one and suppress its activity in the other, whereas in other classes, cerebellar cortical neurons suppress and facilitate their activity to one of the two epochs exclusively. Many of these neurons are simultaneously recorded within sessions and their activity profiles often mirror each other. Here we propose that the patterns observed in these diverse classes can be accounted for by a motif model of the cerebellar cortex that is inspired by recent anatomical discoveries (Witter et al 2016, Arlt & Hausser 2020). We have now better explained these mixed and multiplexed task representations and functional diversities in the main text.

With apologies to the authors, I find these claims rather a stretch and an overly dressed-up presentation of things already known. First, the observation that PCs displayed heterogeneous responses: of course they did. It's well established, from work using several different cerebellar-dependent behaviors, that the responses of PCs are related to the circumstances that activate their climbing fiber input. This has been shown clearly and quantitatively for VOR adaptation, for saccade adaptation and even for eyeblink conditioning. Absent this identification we don't know what output or response the PCs control, and thus we might expect a wide variety of responses with most synchronized to inputs (such as lights and air puffs) rather than to responses.

Our understanding of the reviewer's comment is that it is obvious that the Purkinje cells exhibited heterogeneous response profiles owing to heterogeneity in climbing fiber responses and we ought to have only isolated and analyzed Purkinje cells with an appropriate CF tuning. In the case of eyeblink conditioning, historically, eyeblink-related Purkinje cells have been identified by a US (airpuff) -related complex spike response and it has been claimed (including by our lab) that their simple spike responses often but not always exhibit suppressing responses (Ten Brinke et al 2015, Heiney et al 2014). In the field, it has been quite straightforward to explain the function of suppressing simple spike activity of Purkinje cells but the presence of the facilitating Purkinje cells remains enigmatic and has eluded much discussion let alone a mechanistic explanation.

Here we propose that it is important to look at local Purkinje cell and MLI populations as a whole, and study their interactions because, given recent discoveries on sprawling collateral connectivity of Purkinje cells and disinhibitory motifs in the molecular layer, a departure is needed from the classical view that a population of 40-50 identically-responding Purkinje cells with the same complex spike tuning neatly impinge on a downstream neuron. We make the recommendation that cerebellar computations perhaps need to be re-evaluated through the lens of local motif interactions and how these could give rise to emergent population responses that drive a coherent downstream response.

Finally, the reviewer suggests that we ought to expect responses to be synchronized to inputs such as lights and airpuffs and that these activity patterns are unlikely to reflect eyeblink responses. This could explain a small proportion of our observed data. But passive synchrony to the light and airpuff (each of which lasts tens of milliseconds) would not be able to account for the hundreds of milliseconds of latencies and timescales over which the observed neural activity unfolds after the onset of these stimuli.

These temporal patterns do, however, match the timescales of the eyeblink responses. This point, as well as those highlighted above, are now better explained in the main text.

Second, the computational model the authors present represents in a primitive way processing within a parasagittal stripe. Thus, their “predictions” are only relevant to heterogeneity of responding from PCs within the same stripe, and thus those whose climbing fiber inputs respond to similar stimuli. The model is dressed up with a bit of math, but it is a fairly primitive expression of ideas and computational models that have already been published.

- 1) It assume granule cells fire at different times during a CS
- 2) PCs use LTD/P to learn responses from eyelid conditioning training
- 3) PC-PC and PC-MLI interactions provide ways for PCs to have different responses, which is rather obvious given that in the model PCs can inhibit other PCs.

We do not explicitly claim that the processing in the model lies exclusively within a sagittal stripe. Previous anatomical work has demonstrated that collateralization of PCs in lobule VI is more complex and is not always limited to the parasagittal zone (Witter et al 2016). We do, in fact, model disinhibitory interactions among PCs with different climbing fiber responses. This is now better highlighted in the text.

The reviewer is correct in pointing out that the model’s basis set and plasticity mechanisms are based on previously published models. The novelty of the proposed model lies in the addition of richer granule cell representations to different epochs, greater diversity in climbing fiber activation patterns, and the evaluation of activity emerging from the interaction of cerebellar cortical neurons within local motif architectures.

In sum, because previous work has shown homogeneity of responding within a stripe during eyeblink and other tasks, and given that the authors have not sorted or categorized their PCs according to the climbing fiber responses, the heterogeneity appears to be the result of recording from many PCs that have various and different jobs to do. The computational model is designed to explain heterogeneity within a parasagittal stripe, which is contradicted by previous work and is not demonstrated here. In addition, the primitive nature of the model quite belies the idea that it can have much predictive value anyway.

We are unaware of any experimental claim of homogeneity of responses exclusively within parasagittal stripes during eyeblink conditioning. What has been established is that PCs facilitate and suppress their activity during eyeblink conditioning (Heiney et al 2014, Ten Brinke et al 2015, Halverson et al 2015). Here we elucidate how much more complex and rich the heterogeneity is in swathes of simultaneously recorded neurons. While limited in scope, we believe that a model that attempts to make sense of cerebellar cortical heterogeneity through the lens of recently discovered feedforward and recurrent physiological motifs may add some value to the field. We try to make this point clearer in the text.